# Impact of COVID-19 Pandemic on Thyroid Surgery in a University Hospital in South Korea

**DOI:** 10.3390/cancers14174338

**Published:** 2022-09-05

**Authors:** Seong Hoon Kim, Euna Min, Young Mi Hwang, Yun Suk Choi, Jin Wook Yi

**Affiliations:** Department of Surgery, Inha University Hospital, College of Medicine, Incheon 22332, Korea

**Keywords:** COVID-19, SARS-CoV-2, thyroid nodules, thyroid cancers

## Abstract

**Simple Summary:**

The COVID-19 pandemic has changed healthcare systems around the world. Medical staff focused on managing infectious diseases and other treatments were postponed. We analyzed any changes in the treatment of thyroid cancer during the corona pandemic. Since the outbreak of COVID-19, the number of annual outpatients has significantly decreased, both for new patients and for follow-up patients, and the number of days from the first visit to surgery has increased significantly. After the onset of COVID-19, poor prognostic factors for thyroid cancer increased, and the size of follicular tumors increased.

**Abstract:**

The COVID-19 pandemic has changed healthcare systems around the world. Medical personnel concentrated on infectious disease management and treatments for non-emergency diseases and scheduled surgeries were delayed. We aimed to investigate the change in the severity of thyroid cancer before and after the outbreak of COVID-19 in Korea. We collected three years of data (2019, 2020, and 2021) on patients who received thyroid surgery in a university hospital in South Korea and grouped them as “Before COVID-19”, “After COVID-19 1-year” and “After COVID-19 2-years”. The total number of annual outpatients declined significantly after the outbreak of COVID-19 in both new (1303, 939, and 1098 patients) and follow-up patients (5584, 4609, and 4739 patients). Clinical characteristics, including age, sex, BMI, preoperative cytology results, surgical extent, and final pathologic diagnosis, were not significantly changed after the outbreak of COVID-19. However, the number of days from the first visit to surgery was significantly increased (38.3 ± 32.2, 58.3 ± 105.2, 47.8 ± 124.7 days, *p* = 0.027). Papillary thyroid carcinoma (PTC) patients showed increased proportions of extrathyroidal extension, lymphatic invasion, vascular invasion, and cervical lymph node metastasis. Increased tumor size was observed in patients with follicular tumor (3.5 ± 2.2, 4.0 ± 1.9, 4.3 ± 2.3 cm, *p* = 0.019). After the COVID-19 outbreak, poor prognostic factors for thyroid cancer increased, and an increase in the size of follicular tumors was observed. Due to our study being confined to a single tertiary institution in Incheon city, Korea, nationwide studies that include primary clinics should be required to identify the actual impact of COVID-19 on thyroid disease treatment.

## 1. Introduction

At the end of 2019, atypical pneumonia caused by a new respiratory virus (SARS-CoV-2) was reported in China [1]. This highly contagious virus led to COVID-19, a disease that spread around the world in just two to three months, causing healthcare systems around the world to face an unprecedented crisis. The World Health Organization (WHO) declared COVID-19 a global pandemic on 11 March 2020 [2]. In the early period of the COVID-19 outbreak, the number of COVID-19-infected patients surged, but little was known about this disease; policies were not in place, nor were treatment plans, medications, or vaccines. As such, most healthcare resources, including doctors, nurses, and medical facilities, focused on treating COVID-19-infected patients rather than treating other patients [3]. In the surgery field, specific guidelines were established that postponed elective operations such as oncologic surgeries [4,5,6,7].

Thyroid cancer is one of the most common endocrine malignancies worldwide. The most recent WHO report estimated that the number of new cases of thyroid cancer in 2020 was approximately 449,000 in women and 137,000 in men [8]. The diagnosis of thyroid cancer is based on ultrasound-guided aspiration cytology, mostly during health screening. However, the diagnosis of thyroid cancer was delayed and diminished during the COVID-19 pandemic due to changed healthcare policies, lockdown, and social distancing recommendations [9,10]. In many hospitals, the treatment of thyroid cancer, either surgery or radioactive iodine therapy, was also postponed due to the risk of virus transmission and the redeployment of medical staff [11].

Since thyroid cancer has a good prognosis compared to other solid organ cancers, recent guidelines recommended that diagnosis and thyroid cancer surgery could be delayed during the COVID-19 pandemic [11,12,13,14,15]. However, due to delayed diagnosis and treatment during the pandemic, thyroid cancer tended to progress to worse pathologic status, such as extrathyroidal extension and lymph node metastasis, even if most thyroid cancers typically exhibited indolent behavior. Currently, few studies worldwide have analyzed the changes in the oncologic features of thyroid cancer before and after the outbreak of COVID-19 [16,17,18]. Our aim was to analyze the clinical and pathologic characteristics of thyroid nodules and cancer patients before and after the outbreak of COVID-19 in a single university hospital in Incheon, Korea.

## 2. Methods

In this hospital, a single endocrine surgeon (corresponding author) has performed thyroid surgery since September 2018. For this reason, we reviewed the electronic medical records of the institution for patients with thyroid disease from January 2019 to December 2021. Officially, in South Korea, the first COVID-19 patient was reported on 20 January 2020 [19]. We divided patients into three analysis groups as follows: “Before COVID-19 1-year” from January 2019 to December 2019, “After COVID-19 1-year” from January 2020 to December 2020, and “After COVID-19 2-years” from January 2021 to December 2021. Monthly numbers of COVID-19 diagnosed patients were obtained from the open statistical data at the Korean Center for Disease Control & Prevention (CDC) (http://ncov.mohw.go.kr/en/ (accessed on 1 May 2022)).

With the hospital’s outpatient statistical data, the number of new patients per month and the number of follow-up patients were calculated. We also analyzed patients who received thyroid surgery in our hospital. Patients’ clinical information, such as age, sex, body mass index (BMI, kg/m^2^), postoperative hospital admission days, days from patients’ first outpatient visit to their surgery date, Bethesda cytology results and surgical extent, including thyroidectomy (lobectomy or total thyroidectomy) and lymph node dissection (central or lateral nodes), and the receipt of radioactive iodine therapy (RAI), were reviewed. Pathologic reports, including pathologic diagnosis, largest tumor size, extrathyroidal extension, lymphatic and vascular invasion, lymph node metastasis, and the somatic mutation of *BRAF*^V600E^ and *TERT* promotors, were also retrospectively analyzed. Separate analyses were performed between PTC and follicular-type tumors (benign nodule, noninvasive follicular thyroid neoplasm with papillary-like nuclear features (NIFTP), Hurthle cell carcinoma, follicular thyroid carcinoma, well-differentiated thyroid tumor of uncertain malignant potential (WDT-UMP) and lymphoma).

All of the surgeries were performed by a single endocrine surgeon (J.W.Yi), and the surgical extent was based on the 2015 American Association of Thyroid guideline. Pathologic diagnosis was confirmed by the five pathologists in the authors’ hospital. During the study period, the policy of surgery and pathologic description were not changed. 

The chi-square test was used for the outpatient number analysis. Pearson’s product-moment correlation was used to identify the correlation between the number of COVID-19 patients and the number of thyroid outpatients. For the group comparison, ANOVA was applied to continuous variables, and the chi-square test was used for cross-table analysis. All of the statistical analyses were performed by R-programming language version 4.2.0 [20]. The ethics of this study were approved by the hospital’s institutional review board (IRB number: 2022-06-025).

## 3. Results

Figure 1 shows the number of outpatients in the hospital during the study period. The total number of outpatients, follow-up patients, and new patients was significantly decreased in the “After COVID-19 1-year” and “After COVID-19 2-year” groups compared to the “Before COVID-19” group (chi-square *p* = 0.007). Figure 2A shows the monthly number of confirmed COVID-19 cases in Korea. Due to the community transmission of the omicron variant, the number of COVID-19 patients has markedly increased since November 2021 [21]. The correlation between the number of COVID-19 and thyroid outpatients was not significant during the “After COVID-19 1-year” period (correlation coefficient = −0.170, *p* = 0.598). However, there was a positive correlation during the “After COVID-19 2-year” period (correlation coefficient = 0.668, *p* = 0.017), as illustrated in Figure 2B. Detailed numbers of monthly patients and COVID-19 patients in Korea are listed in Appendix A.

The clinical characteristics of the patients who received thyroid surgery in our hospital are described in Table 1. Patient age, sex, and BMI were not significantly different among the three groups. The preoperative cytology results according to the Bethesda category were also not significantly different among the three groups. A total of 66 to 70% of the patients in the “Before COVID-19”, “After COVID-19 1-year”, and “After COVID-19 2-year” groups were diagnosed as category V and VI (68.2%, 66.6%, and 70.4%, respectively). The surgical extent for thyroidectomy and the surgical extent of lymph node dissection also showed no difference among the three groups. PTC was the most common diagnosis, accounting for more than 70% of cases, followed by follicular tumors in approximately 25% of cases. Postoperative hospital stay days were significantly shorter in the “After COVID-19 1-year” and “After COVID-19 2-year” groups than in the “Before COVID-19” group (3.4 ± 1.1, 3.1 ± 1.0, 3.0 ± 1.4 days, *p* < 0.001, post hoc analysis *p* = 0.002 in “Before COVID-19” vs. “After COVID-19 1-year”, *p* < 0.001 in “Before COVID-19” vs. “After COVID-19 2-year”, *p* = 0.459 in “After COVID-19 1-year” vs. “After COVID-19 2-years”, respectively). The days from the first outpatient visit to surgery were significantly longer in the “After COVID-19 1-year” and “After COVID-19 2-years” groups (42.0 ± 42.7, 62.5 ± 113.1, 57.1 ± 141.8 days, *p* = 0.009, post hoc analysis *p* = 0.009 in “Before COVID-19” vs. “After COVID-19 1-year”, *p* = 0.076 in “Before COVID-19” vs. “After COVID-19 2-year”, *p* = 0.736 in “After COVID-19 1-year” vs. “After COVID-19 2-years”, respectively).

Table 2 shows the pathologic details of the PTC patients. Age, sex, and BMI were not different among the groups. Hospital stay days were significantly shorter in the “After COVID-19” periods than in the “Before COVID-19” periods. The days from the first visit to surgery were significantly longer in the “After COVID-19” periods. Tumor size was not different among the three groups. Furthermore, the proportion of papillary microcarcinoma (PTMC) under 1 cm in diameter was not significantly changed during the three years. However, the proportion of extrathyroidal extension (including microscopic and gross) and the presence of lymphatic invasion and vascular invasion was significantly higher in the “After COVID-19 1-year” and “After COVID-19 2-year” groups. Neck lymph node metastasis for central (N1a) and lateral (N1b) was also significantly frequent in the COVID-19 groups, as listed in Table 2. Somatic mutations of the *BRAF*^V600E^ and *TERT* promotors did not differ among the analyzed groups, nor did the receipt of radioactive iodine therapy.

The details of the follicular tumors are listed in Table 3. Age, sex, and BMI were not different. However, tumor size was significantly increased in the “After COVID-19 1-year” and “After COVID-19 2-year” groups compared to the “Before COVID-19” group (3.5 ± 2.2, 4.0 ± 1.9 and 4.3 ± 2.3 cm, *p* = 0.022, post hoc analysis *p* = 0.217 in “Before COVID-19” vs. “After COVID-19 1-year”, *p* < 0.018 in “Before COVID-19” vs. “After COVID-19 2-year”, *p* = 0.575 in “After COVID-19 1-year” vs. “After COVID-19 2-years”, respectively). Hospital stay days were also shorter in the after-COVID-19 groups. However, days from the first visit to surgery showed no significant difference among the three groups. The final pathologic diagnosis was mostly benign goiters (approximately 90%).

## 4. Discussion

After the global spread of COVID-19, lifestyles worldwide markedly changed. In particular, strict social distancing and lockdown policies in many countries affected general medical and mental health care [22,23,24,25]. In general, health care workers focused on the management of COVID-19 patients due to the outstanding numbers of coronavirus-infected patients [3,7,26,27]. Many surgical societies announced guidelines for elective surgery [4,5,6,28]. They suggested that short-term delays of some elective surgeries were acceptable in the pandemic situation because the delays would make no significant difference in some cancers, including gastric cancer, melanoma, pancreatic cancer, genitourinary cancer, and differentiated thyroid cancer [28].

Thyroid cancer is one of the most important diseases requiring surgical treatment. Although the mortality rate is less than 1 per 100,000, delayed treatment can lead to more extended surgery, including total thyroidectomy, neck dissection, and radioactive iodine therapy, in advanced cases [8]. The patients who received extended surgery and radioactive iodine therapy generally showed no significant difference in recurrence and survival compared to patients who received thyroid lobectomy for PTMC [29,30,31]. However, extended surgery may associate with complications such as an increased rate of vocal cord palsy, hypoparathyroidism, neck discomfort, and large neck scarring. These complications are not life-threatening conditions, but they have harmful effects on the quality of life [32,33,34]. Therefore, the recent American Thyroid Association guidelines recommend considering lobectomy in patients with PTC under 4 cm without clinical evidence of gross extension or lymph node metastasis [31].

During the COVID-19 pandemic, the diagnosis and surgery for thyroid nodules and cancer were inevitably delayed due to the social distancing status and the redeployment of medical resources [10,14,16,17]. Many authoritative thyroid societies suggested that elective thyroid surgery should be postponed, except for urgent conditions such as thyroid nodules or cancer with life-threatening size and organ invasion, rapid growth, and distant metastasis [11,13,14,15]. However, there is limited clinical evidence regarding whether this delayed surgery for thyroid cancers is associated with worse prognostic factors. According to reports from Italy and Jordan, there was no significant difference in the oncological severity of thyroid cancer patients before and after the outbreak of COVID-19 [16,35]. However, a study in China reported that the poor prognostic factors of thyroid cancer, including extrathyroidal extension, multiple tumors, and lymph node metastasis, increased after the COVID-19 outbreak [18]. In a survey study conducted with clinicians majoring in thyroid disease, 87.9% were worried about delayed treatment for their patients during the pandemic [36]. For this reason, more research is needed on how thyroid cancer is being treated after COVID-19.

In addition to delays in diagnosis and treatment caused by the COVID-19 pandemic, it has been reported that COVID-19 virus infection causes genetic and pathological changes to the thyroid gland. The SARS-CoV-2 virus can easily bind to the angiotensin-converting enzyme-2 (ACE2) receptor in human cell membranes, using their spike glycoprotein protein on the external surface of the viral envelope. After the binding process to the ACE2 receptor, receptor transmembrane protease serine 2 (TMPRSS2) cuts the spike protein complex, and the virus can invade the host cells, damaging them. ACE2 protein expression is mainly expressed in the respiratory epithelial cells but is also profoundly expressed in thyroid cells [37]. TMPRSS2 also shows high mRNA expression levels in the thyroid tissue [38]. As so, the thyroid gland is also vulnerable to SARS-CoV-2 infection due to the high expression of ACE2 and TMPRSS2. A histopathologic study using the thyroid glands from the patients who were infected by COVID-19 showed that severe destruction to the parafollicular and follicular epithelial cells leads to the rupture of follicles [39]. Consequently, infections with the SARS-CoV-2 virus are associated with inflammatory thyroid diseases such as subacute thyroiditis, Graves’ disease, thyrotoxicosis, and Hashimoto’s thyroiditis [37]. Increased inflammatory diseases in thyroid glands may be associated with the development of thyroid cancers, but there is no study on whether SARS-CoV-2 infection is directly associated with thyroid cancer development. More prospective research should be performed to find the association between SARS-CoV-2 and thyroid carcinogenesis. 

We performed a detailed analysis of the thyroid patients treated at our hospital for three years, before COVID-19, 1-year after, and 2-years after. During the study period, the number of new patients and follow-up patients was significantly diminished after the COVID-19 pandemic, as shown in Figure 1. Because the study hospital is a tertiary referral hospital, fewer outpatients imply decreased numbers of health screenings in primary care clinics. The correlation between the number of national COVID-19 diagnosed patients and outpatients numbers was not statistically significant in the “After COVID-19 1-year” period but was positive in the “After COVID-19 2-year” period. This phenomenon is thought to show that as the COVID-19 pandemic continues, social distancing is relaxing, and general medical care is being restored.

The annual numbers of thyroid surgeries in 2019, 2020, and 2021 also decreased after the outbreak of COVID-19 (490, 428, and 432, respectively). However, the demographic characteristics of patients, such as age, sex, BMI, and cytologic diagnosis, did not change, as shown in Table 1. Interestingly, the days from the first outpatient visit to surgery in “After COVID-19 1-year” and “After COVID-19 2-year” were significantly prolonged compared to “Before COVID-19” (42.0 ± 42.7, 62.5 ± 113.1, 57.1 ± 141.8 days, *p* = 0.009, respectively). A significant delay was also observed in the PTC patient groups in 2019, 2020, and 2021 (38.3 ± 32.2, 58.3 ± 105.2, 47.8 ± 124.7 days, *p* = 0.027, respectively) but not in the follicular tumor groups. This phenomenon may be partly because thyroid cancer patients were reluctant to have surgery early during the pandemic, and it also seems that the medical staff could not recommend immediate surgery for PTC patients due to COVID-19 precautions. In the case of follicular tumors, tumor size was significantly increased in the “After COVID-19 1-year” and “After COVID-19 2-year” groups compared to the “Before COVID-19” group (3.5 ± 2.2, 4.0 ± 1.9, 4.3 ± 2.3 cm, *p* = 0.022, respectively). Increased tumor size may be due to the delayed diagnosis during the pandemic, but it may have clinical symptoms and may result in prompt surgery without significant delay, as shown in Table 3.

During the study period in 2020 and 2021, our hospital strictly followed its own guidelines for elective surgery during the COVID-19 pandemic. The guideline included that all elective surgery should be performed after the confirmation of negative COVID-19 PCR results within 24-h before the surgery. For the patients who were diagnosed with COVID-19 infection, non-emergency surgery should be delayed until after the negative conversion of the COVID-19 PCR results. As so, our study can safely suggest the effect of delayed diagnosis and treatment on thyroid cancer or thyroid nodule patients without the direct effect of active COVID-19 infection. The most important finding in our study is that, although the proportion of surgical extent for the thyroid gland and lymph node dissection was not significantly changed during the study period, and the frequency of PTMC showed no significant change, the proportion of worse prognostic factors (extrathyroidal extension, lymphatic invasion, vascular invasion, and neck lymph node metastasis, such as N1a and N1b) was significantly increased in PTC patients after the COVID-19 outbreak (Table 2). Perhaps this is due to delayed diagnosis or treatment, but it could also be due to the adverse effect of COVID-19 on the thyroid gland because there are many reports that coronavirus may be associated with Graves’ disease, subacute thyroiditis, or autoimmune thyroid disease [37,40,41,42]. Currently, it is not possible to establish a direct reason why the poor prognostic factors increased in our patients. Related studies that include a history of COVID-19 infection or vaccination and presumptive thyroid disorder will be needed. Additionally, detailed long-term follow-up should be required for patients in the “After COVID-19” group who have the worst prognostic factors. Despite the increase in worse prognostic factors for PTC patients, postoperative admission days were significantly shorter in “After COVID-19 1-year” and “After COVID-19 2-year” in patients with both PTC and follicular tumor. This is because many severe COVID-19 patients are hospitalized in general hospitals, so patients do not want to stay in the hospital for a long time.

The limitations of this study are as follows. This study was only conducted in a single tertiary referral hospital, Incheon, South Korea. As we found a meaningful change after the COVID-19 pandemic in thyroid surgery patients, it cannot reflect the whole national situation. A nationwide study will be performed to evaluate the association between COVID-19 and thyroid disease treatment in the future. Secondly, we suggested that the decreased outpatient numbers in our hospital may be associated with the diminished numbers of health screenings in primary health care clinics due to the COVID-19 pandemic situation. However, we cannot safely conclude this matter without national-scale studies from primary medical clinics. The next limitation is that our patients all had negative results for a COVID-19 PCR test before surgery, and we cannot find the direct result of COVID-19 infection on the severity of the thyroid nodules or cancers. 

Our study is meaningful because it is the first study in Korea that compared thyroid cancer patients before and after the COVID-19 pandemic. Since all outpatient care and surgery were performed by one endocrine surgeon, consistency in treatment policy and surgery pattern was guaranteed. However, since this study was conducted only at a tertiary referral hospital, there is a limitation in that it does not reflect the national status of thyroid patient care. Multicenter and nationwide studies using the National Health Insurance data will be needed to clarify the effect of the pandemic on thyroid disease patients.

## 5. Conclusions

After the outbreak of COVID-19, the number of thyroid outpatients and surgeries decreased. Hospital stay days were reduced, and the time from the first outpatient visit to surgery was prolonged. An increased proportion of worse prognostic factors in PTC and enlarged tumor size in follicular tumors were observed. A nationwide study that includes primary clinics should be performed to support our findings. 

## Figures and Tables

**Figure 1 cancers-14-04338-f001:**
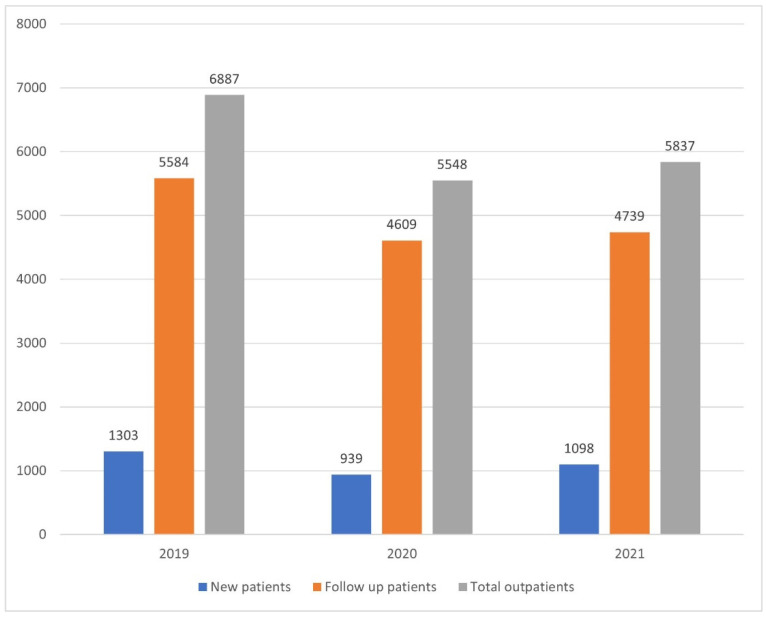
Annual numbers of outpatients during the study period.

**Figure 2 cancers-14-04338-f002:**
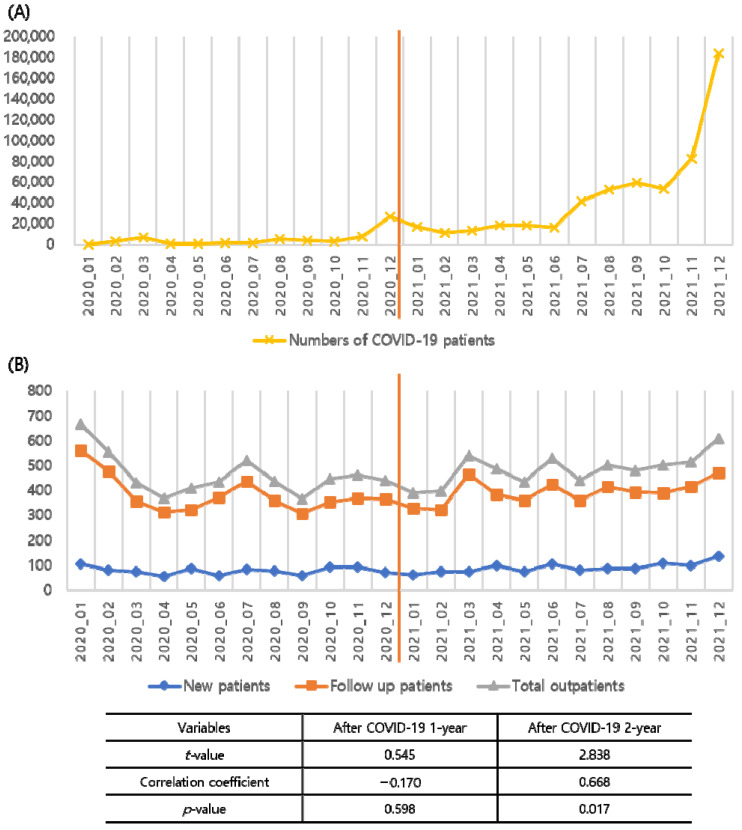
Monthly numbers of COVID-19 diagnosed patients in Korea (**A**) and monthly outpatient numbers in author’s hospital (**B**).

**Table 1 cancers-14-04338-t001:** Clinical characteristics of patients.

Variables	Before COVID-19 1-Year (n = 490)	After COVID-19 1-Year (n = 428)	After COVID-19 2 Year (n = 432)	*p* Value
Age (years, mean ± sd)	49.0 ± 13.2	49.1 ± 13.8	49.2 ± 13.4	0.925
Gender				0.979
Male	105 (35.8%)	93 (21.7%)	95 (22.0%)	
Female	385 (78.6%)	335 (78.3%)	337 (78.0%)	
BMI (kg/m^2^, mean ± sd)	25.1 ± 4.2	25.4 ± 4.1	25.3 ± 4.0	0.652
Postoperative hospital stay days (mean ± sd)	3.4 ± 1.1	3.1 ± 1.0	3.0 ± 1.4	<0.001
Days to 1st visit and surgery (mean ± sd)	42.0 ± 42.7	62.5 ± 113.1	57.1 ± 141.8	0.009
Bethesda category				0.062
I	3 (0.6%)	5 (1.2%)	1 (0.2%)	
II	75 (15.3%)	85 (19.9%)	60 (13.9%)	
III	33 (6.7%)	30 (7.0%)	34 (7.9%)	
IV	37 (7.6%)	16 (3.7%)	30 (6.9%)	
V	80 (16.3%)	67 (15.7%)	89 (20.6%)	
VI	254 (51.8%)	218 (50.9%)	215 (49.8%)	
Completed surgery	8 (1.6%)	7 (1.6%)	3 (0.7%)	
Surgery extent, thyroid				0.098
Lobectomy	249 (52.0%)	236 (56.1%)	251 (59.1%)	
Total thyroidectomy	230 (48.0%)	185 (43.9%)	174 (40.9%)	
Surgery extent, node dissection				0.6
Central	323 (89.2%)	288 (90.6%)	289 (88.1%)	
Lateral	39 (10.8%)	30 (9.4%)	39 (11.9%)	
Pathologic diagnosis				0.141
Papillary thyroid cancer	346 (70.6%)	310 (72.4%)	321 (74.3%)	
Follicular tumors *	125 (25.5%)	105 (24.5%)	104 (24.1%)	
Medullary thyroid cancer	5 (1.0%)	1 (0.2%)	0 (0%)	
Completion surgery	3 (0.6%)	5 (1.2%)	0 (0%)	
Metastatic neck node	11 (2.2%)	7 (1.6%)	7 (1.6%)	

* Benign nodule, noninvasive follicular thyroid neoplasm with papillary-like nuclear features (NIFTP), Hurthle cell carcinoma, follicular thyroid carcinoma, well-differentiated thyroid tumor of uncertain malignant potential (WDT-UMP), and lymphoma.

**Table 2 cancers-14-04338-t002:** Clinical and pathologic characteristics of papillary thyroid cancer patients.

Variables	Before COVID-19, 1-Year (n = 346)	After COVID-19, 1-Year (n = 310)	After COVID-19, 2-Year (n = 321)	*p* Value
Age (years, mean ± sd)	47.4 ± 12.6	47.4 ± 13.4	47.9 ± 12.6	0.862
Gender				0.479
Male	70 (20.2%)	73 (23.5%)	76 (23.7%)	
Female	276 (79.8%)	237 (76.5%)	245 (76.3%)	
BMI (kg/m^2^, mean ± sd)	25.2 ± 4.1	25.3 ± 4.1	25.5 ± 4.3	0.757
Postoperative hospital stay days (mean ± sd)	3.5 ± 1.1	3.2 ± 1.0	3.1 ± 1.6	<0.001
Days to 1st visit and surgery (mean ± sd)	38.3 ± 32.2	58.3 ± 105.2	47.8 ± 124.7	0.027
Tumor size (cm, mean ± sd)	1.0 ± 0.8	1.1 ± 1.0	1.0 ± 0.9	0.405
≤1 cm	232 (67.1%)	203 (65.5%)	224 (69.8%)	0.505
>1 cm	114 (32.9%)	107 (34.5%)	97 (30.2%)	
Multifocality				0.853
Single	208 (60.1%)	181 (58.4%)	194 (60.4%)	
Multiple	139 (39.9%)	129 (41.6%)	127 (39.6%)	
Extrathyroidal extension				<0.001
No	257 (74.2%)	215 (69.4%)	187 (58.3%)	
Yes	89 (25.7%)	95 (30.6%)	134 (41.7%)	
Lymphatic invasion				0.006
Absent	234 (67.8%)	215 (69.6%)	178 (56.7%)	
Indeterminate	35 (10.1%)	33 (10.7%)	50 (15.9%)	
Present	76 (22.0%)	61 (19.7%)	86 (27.4%)	
Vascular invasion				0.002
Absent	307 (89.0%)	269 (87.9%)	249 (79.8%)	
Indeterminate	35 (10.1%)	28 (9.2%)	48 (15.4%)	
Present	3 (0.9%)	9 (2.9%)	15 (4.8%)	
Node stage				0.001
N0, Nx	205 (59.2%)	154 (49.7%)	132 (41.1%)	
N1a	113 (32.7%)	132 (42.6%)	158 (49.2%)	
N1b	28 (8.1%)	24 (7.7%)	31 (9.7%)	
*BRAF* mutation				0.36
Absent	36 (11.3%)	33 (11.2%)	46 (14.5%)	
Present	284 (88.8%)	262 (88.8%)	272 (85.5%)	
*TERT* promotor mutation				0.957
Absent	241 (97.6%)	248 (97.3%)	288 (97.6%)	
Present	6 (2.4%)	7 (2.7%)	7 (2.4%)	
Radioactive iodine therapy				0.126
No	205 (59.2%)	185 (59.7%)	214 (66.7%)	
Yes	141 (40.8%)	125 (40.3%)	107 (33.3%)	

**Table 3 cancers-14-04338-t003:** Clinical and pathologic characteristics of follicular tumors patients.

Variables	Before COVID-19, 1-Year (n = 125)	After COVID-19, 1-Year (n = 105)	After COVID-19, 2-Year (n = 104)	*p* Value
Age (years, mean ± sd)	52.0 ± 14.1	53.8 ± 14.1	53.4 ± 15.0	0.62
Gender				0.582
Male	27 (21.6%)	19 (18.1%)	17 (16.3%)	
Female	98 (78.4%)	86 (81.9%)	87 (83.7%)	
BMI (kg/m^2^, mean ± sd)	25.0 ± 4.5	25.3 ± 3.9	24.9 ± 3.1	0.772
Postoperative hospital stay days (mean ± sd)	3.1 ± 1.0	2.8 ± 0.8	2.9 ± 0.9	0.019
Days to 1st visit and surgery (mean ± sd)	45.8 ± 48.3	63.1 ± 111.0	69.9 ± 154.9	0.23
Tumor size (cm, mean ± sd)	3.5 ± 2.2	4.0 ± 1.9	4.3 ± 2.3	0.022
Multifocality				0.235
Single	104 (83.2%)	93 (88.6%)	94 (90.4%)	
Multiple	21 (16.8%)	12 (11.4%)	20 (9.6%)	
Pathologic diagnosis				n/a
Benign	112 (89.6%)	95 (90.5%)	91 (87.5%)	
NIFTP	11 (8.8%)	6 (5.7%)	6 (5.8%)	
Hurthle cell carcinoma	0	0	5 (4.8%)	
Follicular thyroid carcinoma	2 (1.6%)	2 (1.9%)	0	
WDT-UMP	0	2 (1.9%)	1 (1.0%)	
Lymphoma	0	0	1 (1.0%)	

## Data Availability

The data presented in this study are available in this article and Appendix A.

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
