# Peer review of "Impact of COVID-19 Pandemic on Thyroid Surgery in a University Hospital in South Korea"

_cancers, 2022, doi:10.3390/cancers14174338_

Round 1
Reviewer 1 Report
This manuscript aimed to investigate the change in the severity of thyroid cancer before and after the outbreak of COVID-19 in Korea. Its a clinical survey that has some serious limitations including:
1. Authors didn't discuss the molecular and genomics basis of the relationship between SARS-COV19 with thyroid cancer. They also didn't mention the pathological ground of this relationship.
2. This survey was conducted based on one institution. Data of one institution never reflect the actual scenario.
3. Author themself understand the limitation by mentioning ". During the study period, the numbers of new patients and follow-up patients were significantly diminished after the COVID-19 pandemic, as shown in Figure 1. Because the study hospital is a tertiary referral hospital, fewer outpatients imply decreased numbers of health screenings in primary care clinics. In this regard, there is a need for national-scale studies of primary medical clinics to support our findings."
4. Authors also mentioned "The most important finding in our study is that, although the proportion of surgical extent for the thyroid gland and lymph node dissection was not significantly changed during the study period, the proportion of worse prognostic factors were significantly increased in PTC patients after the COVID-19 outbreak. Perhaps this is due to delayed diagnosis or treatment, but it could also be due to the adverse effect of the coronavirus on the thyroid gland because there are many reports that". However, the authors never mentioned whether these all patients are COVID positive, if so how was their recovery procedures, and what medications they were in. Missing these pieces of information makes these statements weak.
Author Response
This manuscript aimed to investigate the change in the severity of thyroid cancer before and after the outbreak of COVID-19 in Korea. Its a clinical survey that has some serious limitations including:
- Authors didn't discuss the molecular and genomics basis of the relationship between SARS-COV19 with thyroid cancer. They also didn't mention the pathological ground of this relationship.
(Ans)
Thank you for your profound opinion. As you pointed out, there are many studies on the genomic and pathologic association between COVID-19 and thyroid cancer. However, since COVID-19 has only been spread for about three years, there are still few studies that provide solid evidence for this. We summarized the molecular and pathologic relationships, known so far and added it in the discussion section as follows.
(Add)
In addition to delays in diagnosis and treatment caused by COVID-19 pandemic, it has been reported that COVID-19 virus infection causes genetic and pathological changes in the thyroid gland. SARS-CoV-2 virus can easily bind to the angiotensin converting en-zyme-2 (ACE2) receptor in human cell membrane, using their spike glycoprotein protein on the external surface of viral envelope. After the binding process to ACE2 receptor, receptor transmembrane protease serine 2 (TMPRSS2) cuts the spike protein complex and virus can invade the host cells and damaging them. ACE2 protein expression is mainly expressed in the respiratory epithelial cells, but also profoundly expressed in the thyroid cells [37]. TMPRSS2 also shows high mRNA expression level in the thyroid tissue [38]. As so, thyroid gland also vulnerable to the SARS-CoV-2 infection due to the high expression of ACE2 and TMPRSS2. Histopathologic study using the thyroid gland from the patients who infected by COVID-19 showed that severe destruction to the parafollicular and follicular epithelial cells, lead to rupture of follicles [39]. Consequently, infection of SARS-CoV-2 virus are associated with inflammatory thyroid diseases such as subacute thyroiditis, Graves’ disease, thyrotoxicosis, and Hashimoto’s thyroiditis [37]. Increased inflammatory diseases in thyroid glands may associated with the development of thyroid cancers, but there’s no study that SARS-CoV-2 infection is directly associated with thyroid cancer development. More prospective research should be performed to find the association be-tween SARS-CoV-2 and thyroid carcinogenesis.
- This survey was conducted based on one institution. Data of one institution never reflect the actual scenario.
(Ans)
As you pointed out, this study was only conducted in single institution and does not reflect the national situation and actual scenario of thyroid cancer treatment. We added this issue in the discussion section as follows.
(Add)
Limitations of this study are as follows. This study only conducted in the single tertiary referral hospital, Incheon, South Korea. As we found a meaningful change after the COVID-19 pandemic in thyroid surgery patients, it cannot reflect the whole national situation. Nationwide study will be performed to evaluate the association between COVID-19 and thyroid disease treatment in the future.
- Author themself understand the limitation by mentioning ". During the study period, the numbers of new patients and follow-up patients were significantly diminished after the COVID-19 pandemic, as shown in Figure 1. Because the study hospital is a tertiary referral hospital, fewer outpatients imply decreased numbers of health screenings in primary care clinics. In this regard, there is a need for national-scale studies of primary medical clinics to support our findings."
(Ans)
We totally agree with your opinion. According to your suggestion, we modified the discussion little bit, and added this limitation in more detail, as follows.
(Add)
In second, we suggested that decreased outpatients’ numbers in our hospital may associated with the diminished numbers of health screening in primary health care clinics due to the COVID-19 pandemic situation. However, we cannot safely conclude this matter without the national-scale studies from the primary medical clinics.
- Authors also mentioned "The most important finding in our study is that, although the proportion of surgical extent for the thyroid gland and lymph node dissection was not significantly changed during the study period, the proportion of worse prognostic factors were significantly increased in PTC patients after the COVID-19 outbreak. Perhaps this is due to delayed diagnosis or treatment, but it could also be due to the adverse effect of the coronavirus on the thyroid gland because there are many reports that". However, the authors never mentioned whether these all patients are COVID positive, if so how was their recovery procedures, and what medications they were in. Missing these pieces of information makes these statements weak.
(Ans)
You also pointed out one of the important issues for the relationship between COVID-19 infection and thyroid cancer. We totally agree with your consideration. During the study period in 2020 and 2021, our hospital follows strict guidelines for elective surgery. We can only do the surgery for thyroid after the check the negative results for COVID-19 PCR. As so, all patients during the study period are negative in their PCR result. For the patients who infected in the COVID-19 virus before the surgery, they cannot receive their thyroid surgery unless their PCR test converted into negative. As so, our study can suggest that delayed diagnosis and treatment is associated with worsening the thyroid cancer severity and increased tumor diameter in thyroid nodule patients. We added this important issue in the discussion section as follows. Also, this matter discussed again in the limitation section as follows.
(Add)
- During the study period in 2020 and 2021, our hospital strictly follows the own guidelines for elective surgery during the COVID-19 pandemic. The guideline include that all elective surgery should be done after the confirmation of negative COVID-19 PCR results within 24-hours before the surgery. For the patients who diagnosed as COVID-19 infection, non-emergency surgery should be delayed after the negative conversion of COVID-19 PCR results. As so, our study can safely suggest the effect of delayed diagnosis and treatment to the thyroid cancer or thyroid nodule patients, without the direct effect of active COVID-19 infection.
- Next limitation is our patients were all negative results for COVID-19 PCR test before surgery and we cannot find the direct result of COVID-19 infection to the severity of thyroid nodule or cancers.
Reviewer 2 Report
Min et al. aimed to investigate the change in the severity of thyroid cancer before and after the outbreak of COVID-19 in Korea. They collected data from year 2019, 2020 and 2021 on patients who received thyroid surgery in a tertiary center in South Korea. The total numbers of annual outpatients declined significantly after the outbreak of COVID-19 in both new and follow-up patients. Clinical characteristics, including age, sex, BMI, preoperative cytology results, surgical extent and final pathologic diagnosis, were not significantly changed after the outbreak of COVID-19. However, the number of days from the 1st visit to surgery was significantly increased (38.303±32.160, 58.306±105.156, 47.757±124.702 days, p=0.027). Authors report that patients with papillary thyroid carcinoma patients showed increased proportions of extrathyroidal extension, lymphatic invasion, vascular invasion, and cervical lymph node metastasis after outbreak of COVID-19. Increased tumor size was observed in patients with follicular tumor. Min et al.’s manuscript is interesting, however needs further clarification of several topics. Namely, other factors that might influenced results are missing: selection bias because of different patient referral during COVID-19 epidemic and proportion of patients who were offered active surveillance instead of surgical procedure by endocrinologists. My view is that major revision of the manuscript should be done.
Major remarks
Please comment on selection bias because of different patient referral during COVID-19 epidemic. Smaller hospitals in some countries during COVID-19 epidemic did not do thyroid surgery. Who referred patients to your hospital? Are you the only referral center for them or in the region?
How in South Korea changed proportion of patients who were offered active surveillance instead of surgical procedure by endocrinologists during observation period? Authors should present their data in Table 1 and Table 2 with regard to microcarcinoma (≤ 1 cm versus >1cm).
It seems very unlikely that so short delay of surgery (38.303±32.160, 58.306±105.156, 47.757±124.702 days, p=0.027) for patients with such small papillary carcinoma or even microcarcinoma have such an impact on extrathyroidal extension, lymphatic invasion, vascular invasion, and cervical lymph node metastasis.
Who did pathology exams? Did they change their protocols?
Why you have such long postoperative hospital stay? Reimbursement?
Paragraph about limitations is missing
Because of all before mentioned concerns, I believe that the last sentence of the conclusion section (lines 259-261) should be omitted.
Minor remarks:
Abstract: »Papillary thyroid carcinoma (PTC) patients showed increased proportions of extrathyroidal extension, lymphatic invasion, vascular invasion, and cervical lymph node metastasis (40.751%, 50.323% and 58.878%, p<0.001), respectively.« How it comes? This sentence is not precise. Please omit the numbers.
Abstract: »More delicate management of thyroid cancer patients is needed.« This is not a proper conclusion.
Line 120: »area of lymph node dissection«. What is area of lymph node disection?
Lines 143-146. Was the proportion of patients with microcarcinoma the same in all three years?
Line 127: Reference #28 is inappropriate and unnecessary
Lines 179-182 This statement is true for microcarcinoma!
Liine:182: »extended surgery is inevitably associated with complications«. It may be associated.
Line 187: »[34].« The correct reference is 31. Reference »34 is irrelevant in this context.
Author Response
Major remarks
Please comment on selection bias because of different patient referral during COVID-19 epidemic. Smaller hospitals in some countries during COVID-19 epidemic did not do thyroid surgery. Who referred patients to your hospital? Are you the only referral center for them or in the region?
(Ans)
Thank you for your important point out. Authors’ hospital is located in the Incheon metropolitan city, approximately 3 million population, located 28km from Seoul, the capital of South Korea. This city has only 3 university hospitals and author’s hospital is one of them. Around 3,000 private medical clinics are working in Incheon city and they refer the patients to university hospitals. (https://www.incheon.go.kr/index). As you were mentioned, smaller hospitals didn’t perform thyroid surgery, same in Korea. However, large university hospitals like ours have continued to perform surgeries for many diseases despite the COVID-19 situation. As you pointed out, we think that selection bias may exist in the fact that our study was conducted only with patients in the Incheon area. We added about this in the limitation section as follows.
(Add)
Limitations of this study are as follows. This study only conducted in the single tertiary referral hospital, Incheon, South Korea. As we found a meaningful change after the COVID-19 pandemic in thyroid surgery patients, it cannot reflect the whole national situation. Nationwide study will be performed to evaluate the association between COVID-19 and thyroid disease treatment in the future.
How in South Korea changed proportion of patients who were offered active surveillance instead of surgical procedure by endocrinologists during observation period? Authors should present their data in Table 1 and Table 2 with regard to microcarcinoma (≤ 1 cm versus >1cm).
It seems very unlikely that so short delay of surgery (38.303±32.160, 58.306±105.156, 47.757±124.702 days, p=0.027) for patients with such small papillary carcinoma or even microcarcinoma have such an impact on extrathyroidal extension, lymphatic invasion, vascular invasion, and cervical lymph node metastasis.
(Ans)
Table 1 listed all patients including PTC and follicular tumor patients. Table 2 contains the PTC patients and we added the size classification, ≤ 1 cm versus >1cm in table 2. And we find out that proportion of PTMC are increased in the year of 2021. However, there’s no statistic significance. According to that, although the proportion of PTMC was increased, aggressive cancer indicators such as extrathyroidal extension, lymphatic invasion, vascular invasion, and cervical lymph node metastasis are increased in the period of After COVID-19. We added this factor in the result and discussion section as follows. Thank you for your valuable comment.
(Add)
Results)
And proportion of papillary microcarcinoma (PTMC) under the 1cm of diameter was not significantly changed during the three years.
Discussion)
The most important finding in our study is that, although the proportion of surgical extent for the thyroid gland and lymph node dissection were not significantly changed during the study period, and frequency of PTMC did not showed significant change, the proportion of worse prognostic factors (extrathyroidal extension, lymphatic invasion, vascular invasion, and neck lymph node metastasis, such as N1a and N1b) were significantly increased in PTC patients after the COVID-19 outbreak (table 2).
Who did pathology exams? Did they change their protocols?
(Ans)
Author’s hospital has department of pathology, and we have five professor pathologists. Suring the study period, they continued to follow same pathologic protocols. We added it in the method section as follows.
(Add)
All surgery was done by single endocrine surgeon (J.W.Yi) and surgical extent was based on the 2015 American Association of Thyroid guideline. Pathologic diagnosis was confirmed by the five pathologists in authors’ hospital. During the study period, policy of surgery and pathologic description was not changed.
Why you have such long postoperative hospital stay? Reimbursement?
(Ans)
Unlike other hospitals in the West, most hospitals in Korea recommend inpatient treatment for 2 or 3 days after thyroid surgery. The reason is that in Korea, the national health insurance pays certain expenses for inpatients, and especially for cancer patients, the national health insurance pays 95% of patients’ medical expenses. This is a special situation in Korea, so it is not necessary to mention it in the main text.
Paragraph about limitations is missing
(Ans)
We added the limitation section as follows. Thanks for your comments.
(Add)
Limitations of this study are as follows. This study only conducted in the single tertiary referral hospital, Incheon, South Korea. As we found a meaningful change after the COVID-19 pandemic in thyroid surgery patients, it cannot reflect the whole national situation. Nationwide study will be performed to evaluate the association between COVID-19 and thyroid disease treatment in the future. In second, we suggested that decreased outpatients’ numbers in our hospital may associated with the diminished numbers of health screening in primary health care clinics due to the COVID-19 pandemic situation. How-ever, we cannot safely conclude this matter without the national-scale studies from the primary medical clinics. Next limitation is our patients were all negative results for COVID-19 PCR test before surgery and we cannot find the direct result of COVID-19 infection to the severity of thyroid nodule or cancers.
Because of all before mentioned concerns, I believe that the last sentence of the conclusion section (lines 259-261) should be omitted.
(Ans)
We are agree with your opinion. We omitted the sentences and changed into following sentence.
(Add)
Nationwide study including the primary clinics should be performed to support our findings.
Minor remarks:
Abstract: »Papillary thyroid carcinoma (PTC) patients showed increased proportions of extrathyroidal extension, lymphatic invasion, vascular invasion, and cervical lymph node metastasis (40.751%, 50.323% and 58.878%, p<0.001), respectively.« How it comes? This sentence is not precise. Please omit the numbers.
(Ans)
We deleted the numbers. Thank of your detailed comment.
Abstract: »More delicate management of thyroid cancer patients is needed.« This is not a proper conclusion.
(Ans)
We changed out conclusion as follows. Your comment made our abstract more convincing.
(Before)
More delicate management of thyroid cancer patients is needed.
(After)
Due to our study is confined in the single tertiary institution in Incheon city, Korea, nationwide study including the primary clinics should be required to identify the actual impact of COVID-19 to the thyroid disease treatment.
Line 120: »area of lymph node dissection«. What is area of lymph node disection?
(Ans)
We changed it into the “surgical extent of lymph node dissection”. Thank you for your point out.
Lines 143-146. Was the proportion of patients with microcarcinoma the same in all three years?
(Ans)
Proportion of PTMC was not different during the study period as listed in table 2, and stated in the result section as follows.
(Add)
And proportion of papillary microcarcinoma (PTMC) under the 1cm of diameter was not significantly changed during the three years.
Line 127: Reference #28 is inappropriate and unnecessary
(Ans)
We omitted the reference. Thanks.
Lines 179-182 This statement is true for microcarcinoma!
(Ans)
We changed this sentence as follows. Thank for your correction.
(Before)
Patients who received extended surgery and radioactive iodine therapy generally showed no significant difference in recurrence and survival compared to patients who received thyroid lobectomy for papillary thyroid cancers
(After)
Patients who received extended surgery and radioactive iodine therapy generally showed no significant difference in recurrence and survival compared to patients who received thyroid lobectomy for PTMC
Liine:182: »extended surgery is inevitably associated with complications«. It may be associated.
(Ans)
We changed it as follows.
(Before)
extended surgery is inevitably associated with complications
(After)
extended surgery may associate with complications
Line 187: »[34].« The correct reference is 31. Reference »34 is irrelevant in this context.
(Ans)
We changed the references. Thanks.
Reviewer 3 Report
In this paper the Authors analayzed the Impact of COVID-19 pandemic on thyroid surgery in a university hospital in South Korea. It is a debated and novel topic. A comprehensive and extensive literature review of the NCBI database PubMed was also carried out. The article was well conducted and it is interesting in its fields. It is a well-structured paper, written in good English and the References are up dated.
Minor issues:
In more demolitive surgery, interventions are affected by more severe complications. In the “discussion” section I suggest to better analyze this topic. Therefore, the following paper should be considered:
“Gambardella, C., Patrone, R., Di Capua, F. et al. The role of prophylactic central compartment lymph node dissection in elderly patients with differentiated thyroid cancer: a multicentric study. BMC Surg 18, 110 (2019).”
“Gambardella C, Pagliuca R, Pomilla G, Gambardella A. COVID-19 risk contagion: Organization and procedures in a South Italy geriatric oncology ward. J Geriatr Oncol. 2020 May 22:S1879-4068(20)30237-X. doi: 10.1016/j.jgo.2020.05.008”
Author Response
In this paper the Authors analayzed the Impact of COVID-19 pandemic on thyroid surgery in a university hospital in South Korea. It is a debated and novel topic. A comprehensive and extensive literature review of the NCBI database PubMed was also carried out. The article was well conducted and it is interesting in its fields. It is a well-structured paper, written in good English and the References are up dated.
Minor issues:
In more demolitive surgery, interventions are affected by more severe complications. In the “discussion” section I suggest to better analyze this topic. Therefore, the following paper should be considered:
“Gambardella, C., Patrone, R., Di Capua, F. et al. The role of prophylactic central compartment lymph node dissection in elderly patients with differentiated thyroid cancer: a multicentric study. BMC Surg 18, 110 (2019).”
“Gambardella C, Pagliuca R, Pomilla G, Gambardella A. COVID-19 risk contagion: Organization and procedures in a South Italy geriatric oncology ward. J Geriatr Oncol. 2020 May 22:S1879-4068(20)30237-X. doi: 10.1016/j.jgo.2020.05.008”
(Ans)
Thanks for recommending good references. These references are a good source of support for our opinion. We cited those references additionally on lines 47, 177-179 and 192. (reference 7, 34)
Round 2
Reviewer 1 Report
Dear editor
The authors address my queries properly and I don't have any more observations. The manuscript can be accepted in its current form.
Thank you